# Selective hematopoietic stem cell ablation using CD117-antibody-drug-conjugates enables safe and effective transplantation with immunity preservation

Agnieszka Czechowicz[1,2,3,4,5,6,7], Rahul Palchaudhuri[4,5,8,9,10], Amelia Scheck[1,3,4,5,6,7], Yu Hu [1], Jonathan Hoggatt[4,5,8], Borja Saez[4,5,8,11], Wendy W. Pang[7,12,13,14], Michael K. Mansour[4,5,8,15], Tiffany A. Tate[4,5,8], Yan Yi Chan[6,7], Emily Walck[6,7], Gerlinde Wernig[7,16], Judith A. Shizuru [7,13,14], Florian Winau[1], David T. Scadden[4,5,8] & Derrick J. Rossi [1,3,4,5]

Hematopoietic stem cell transplantation (HSCT) is a curative therapy for blood and immune diseases with potential for many settings beyond current standard-of-care. Broad HSCT application is currently precluded largely due to morbidity and mortality associated with genotoxic irradiation or chemotherapy conditioning. Here we show that a single dose of a CD117-antibody-drug-conjugate (CD117-ADC) to saporin leads to > 99% depletion of host HSCs, enabling rapid and efficient donor hematopoietic cell engraftment. Importantly, CD117-ADC selectively targets hematopoietic stem cells yet does not cause clinically significant side-effects. Blood counts and immune cell function are preserved following CD117-ADC treatment, with effective responses by recipients to both viral and fungal challenges. These results suggest that CD117-ADC-mediated HSCT pre-treatment could serve as a non-myeloablative conditioning strategy for the treatment of a wide range of non-malignant and malignant diseases, and might be especially suited to gene therapy and gene editing settings in which preservation of immunity is desired.

[1] Program in Cellular and Molecular Medicine, Department of Medicine, Boston Children's Hospital, Boston, MA 02115, USA. [2] Department of Pediatric Oncology, Dana Farber Cancer Institute, Boston, MA 02115, USA. [3] Department of Pediatrics, Division of Hematology/Oncology, Harvard Medical School, Boston, MA 02115, USA. [4] Department of Stem Cell and Regenerative Biology, Harvard University, Cambridge, MA 02138, USA. [5] Harvard Stem Cell Institute, Cambridge, MA 02138, USA. [6] Department of Pediatrics, Division of Stem Cell Transplantation and Regenerative Medicine, Stanford University School of Medicine, Stanford, CA 94305, USA. [7] Institute for Stem Cell Biology and Regenerative Medicine, Stanford University School of Medicine, Stanford, CA 94305, USA. [8] Center for Regenerative Medicine, Massachusetts General Hospital, Boston, MA 02114, USA. [9] Department of Chemistry and Chemical Biology, Harvard University, Cambridge, MA 02138, USA. [10] Magenta Therapeutics, Cambridge, MA 02139, USA. [11] Center For Applied Medical Research, Pamplona 31008, Spain. [12] Department of Medicine, Division of Hematology, Stanford University School of Medicine, Stanford, CA 94305, USA. [13] Department of Medicine, Division of Blood and Marrow Transplantation, Stanford University School of Medicine, Stanford, CA 94305, USA. [14] Stanford Cancer Institute, Stanford University School of Medicine, Stanford, CA 94305, USA. [15] Division of Infectious Diseases, Massachusetts General Hospital, Boston, MA 02114, USA. [16] Department of Pathology, Stanford University School of Medicine, Stanford, CA 94305, USA. These authors contributed equally: Agnieszka Czechowicz, Rahul Palchaudhuri. These authors jointly supervised this work: Derrick J. Rossi, David T. Scadden. Correspondence and requests for materials should be addressed to A.C. (email: aneeshka@stanford.edu) or to D.T.S. (email: david_scadden@harvard.edu) or to D.J.R. (email: derrick.rossi@childrens.harvard.edu)

Hematopoietic stem cell transplantation (HSCT) is a powerful treatment modality that enables replacement of host hematopoietic stem cells (HSCs) with HSCs from a healthy donor or genetically improved/corrected HSCs from the patient[1]. This procedure often results in life-long benefits and can curatively treat many malignant and non-malignant blood and immune diseases. Hence >1,000,000 patients have been transplanted in the last 60+ years for a wide range of blood and immune diseases, including leukemias, hemoglobinopathies, metabolic diseases, immunodeficiencies, and even HIV[2]. HSCT has also been demonstrated to be a beneficial treatment for autoimmune diseases[3], and, with modern gene-modification techniques such as lentiviral transduction and ZFN, TALEN, or CRISPR/Cas9 gene editing, HSCT application may be expanded to an even wider range of diseases[4].

However, despite its broad curative potential, HSCT is currently mainly restricted to otherwise incurable malignant diseases and it is estimated that <25% of patients that could benefit from HSCT undergo transplantation[5]. This is largely due to undesirable morbidity/mortality from cytotoxic chemotherapy and irradiation-based conditioning currently necessary to enable donor HSC engraftment and the risks associated with graft versus host disease (GvHD). Due to their non-specific nature, classic conditioning regimens lead to both detrimental short-term and long-term complications including multi-organ damage, mucositis, need for frequent red blood cell and platelet transfusions, infertility, and secondary malignancies[6,7]. Additionally, these agents result in profound and prolonged immune ablation, which predisposes patients to serious and sometimes fatal opportunistic infections necessitating extended hospitalizations and exposure to toxic side effects of anti-infective agents[8]. Although much work has led to the development of reduced intensity conditioning (RIC) methods, which use lower dose combination chemotherapy with or without low dose irradiation, patients still experience many of these debilitating side effects[9]. Eliminating such harsh conditioning regimens would dramatically improve HSCT and expand its use, especially when combined with gene therapy or gene editing where the native hematopoietic system can be repaired without the need for allogeneic transplantation which carries GvHD and immune suppression risk.

Traditionally, conditioning involves total body irradiation (TBI) and/or various chemotherapy prior to HSCT. These agents have been thought essential to make "space" in host bone marrow (BM) for donor HSC engraftment[10], but they are non-specific and induce significant collateral damage. We previously demonstrated in immunodeficient mice that host HSC competition specifically limits donor HSC engraftment[11,12]. Subsequently, we showed that host HSCs in this model could be depleted using an antagonistic anti-murine CD117 monoclonal antibody (ACK2), resulting in an effective, safe, alternative single-agent conditioning approach enabling high donor HSC engraftment[11]. However, this naked antibody conditioning approach only functions as a stand-alone agent in certain disease models; such as immune deficiency[11,13] and Fanconi anemia[14]. In other settings, it has been found necessary to combine ACK2 with agents such as low-dose irradiation[15] or CD47 antagonism[13] to increase potency, making clinical translation of this approach challenging.

We have recently shown that an alternative antibody-based approach to transplant conditioning is through use of CD45.1 or CD45.2 antibodies conjugated to the drug saporin[16]. Saporin is a ribosome-inactivating protein with potent cell-cycle-independent cytotoxic activity[17]. Unlike other toxins, it lacks a general cell entry domain and on its own is non-toxic. It can be targeted to specific cell types by coupling to antibodies directed to various cell-surface antigens and it is believed that upon receptor-mediated internalization, saporin is released intracellularly

halting protein synthesis and inducing cell death[17]. As CD45 is present on most hematopoietic cells, including HSCs, we found CD45-antibody-drug-conjugates (CD45-ADCs) to be effective conditioning agents in various syngeneic immunocompetent mouse models[16]. However, as CD45 is also present on all lymphocytes, CD45-ADCs lead to profound lymphodepletion[16] and therefore likely will maintain opportunistic infection susceptibility. Therefore, if this approach is translated, it may be suited for HSCT contexts where immune depletion is required (e.g. allotransplant and autoimmune disease treatment), but is likely to be suboptimal for other applications. For settings such as autologous gene therapy, an improved solution in which HSCs can be specifically depleted while maintaining intact immunity would be optimal.

Here, we show that CD117 antibody-drug-conjugates (CD117-ADCs) effectively and specifically deplete host HSCs in vivo with minimal toxicity. This allows safe and highly efficient transplantation of immunocompetent mice with whole bone marrow (WBM) or purified HSCs, whereas downstream effector cells are spared due to a lack of CD117 expression, thereby leading to the preservation of immunity.

## Results

**CD117-ADC potently depletes endogenous HSCs.** We created a single-agent antibody-drug-conjugate targeting the antigen CD117 (c-kit), which is a critical hematopoietic stem and progenitor cell receptor for the cytokine stem cell factor (SCF)[18]. Within the hematopoietic compartment, CD117 expression is largely restricted to HSCs and proximal multi-potent and oligo-potent progenitors, though it is also expressed on some downstream hematopoietic effector cells such as mast cells and rare cells of other tissues[19]. The CD117-ADC was prepared by combining biotinylated anti-CD117 antibody (clone 2B8) with streptavidin–saporin (Fig. 1a). In separate experiments, the CD117-ADC and naked CD117 antibody were administered by intravenous injection to immunocompetent, wild-type mice, and HSC depletion was assayed in BM 8 days after injection. HSCs (defined immunophenotypically as Lin−cKit+Sca1+CD150+CD48−) were quantified by flow cytometry of the BM of treated mice (Fig. 1b–d), and functionally by transplantation of the BM into lethally irradiated recipients (Fig. 1e, f). These studies revealed that 1.5 mg/kg of CD117-ADC (~12 µg streptavidin-saporin) optimally resulted in depletion of >99% of immunophenotypic and functional HSCs.

As CD117 is restricted to approximately 5–10% of BM cells, overall BM cellularity remained unchanged with such treatment (Supplementary Fig 1a, b). Surprisingly many CD117-expressing immunophenotypic progenitors remained present at the day 8 timepoint post CD117-ADC treatment (Supplementary Fig 1c), although functional progenitor activity, as assessed by in vitro colony-forming activity, showed continued dose-dependent depletion at this timepoint (Supplementary Fig 1d). Although all CD117-expressing hematopoietic progenitor sub-populations assayed were decreased at early timepoints after CD117-ADC treatment, depletion was to a lesser degree than after sub-lethal 5 Gy TBI and importantly these progenitors rapidly rebounded to near normal levels by day 6 post treatment (Supplementary Fig 1e–j, Supplementary Fig 2).

**CD117-ADC enables efficient donor-cell engraftment.** We next determined whether the HSC depletion achieved by CD117-ADC conditioning enabled successful engraftment of donor hematopoietic cells. Consistent with our previously published study investigating CD45-ADCs[16], ten million congenic CD45.1 or syngeneic CD45.2–GFP WBM cells were used as donor cells for

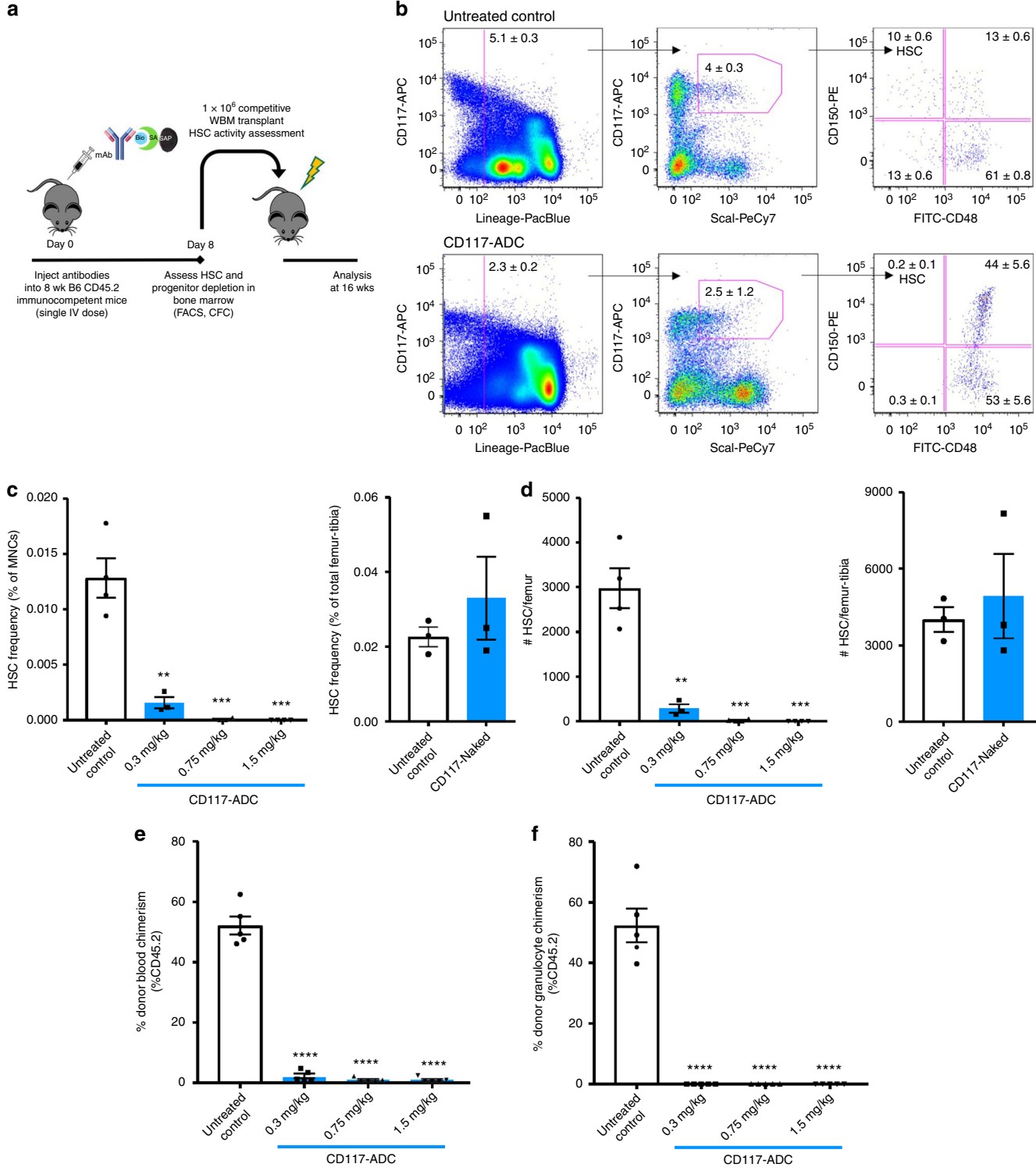

**Fig. 1** One-time, intravenous treatment with CD117-ADC potently depletes hematopoietic stem cells (HSCs). **a** Experimental outline for assessing the ability of antibody-drug-conjugates to deplete HSCs in immunocompetent, wild-type C57BL/6 mice. HSCs and progenitors were assessed through flow cytometric (FACS) phenotypic analysis and colony-forming cell-forming assays, and in vivo reconstitution potential was assessed by competitive transplantation assays. **b** Representative FACS plots of bone marrow from untreated control and CD117-ADC-treated animals (2B8 clone bound to saporin toxin). **c**, **d** Dose-dependent phenotypic depletion of HSCs (Lin−cKit+Sca1+CD150+CD48−) by CD117-ADC and lack thereof by naked CD117 mAb as assessed 8 days after IV administration, displaying decreased HSC frequency (**c**) and total HSC number (**d**). Non-treated mice served as controls. **e**, **f** In vivo depletion of HSCs affirmed by lack of long-term reconstitution activity of treated WBM as assessed by donor total blood chimerism (**e**) and donor granulocyte chimerism (**f**) in competitive transplantation assays. Statistics calculated using unpaired *t* test. Data represent mean ± SEM. (*n* = 3–5 mice/ group, assayed individually); all data points significant as indicated vs. untreated control (**P < 0.01, ***P < 0.001, ****P < 0.0001)

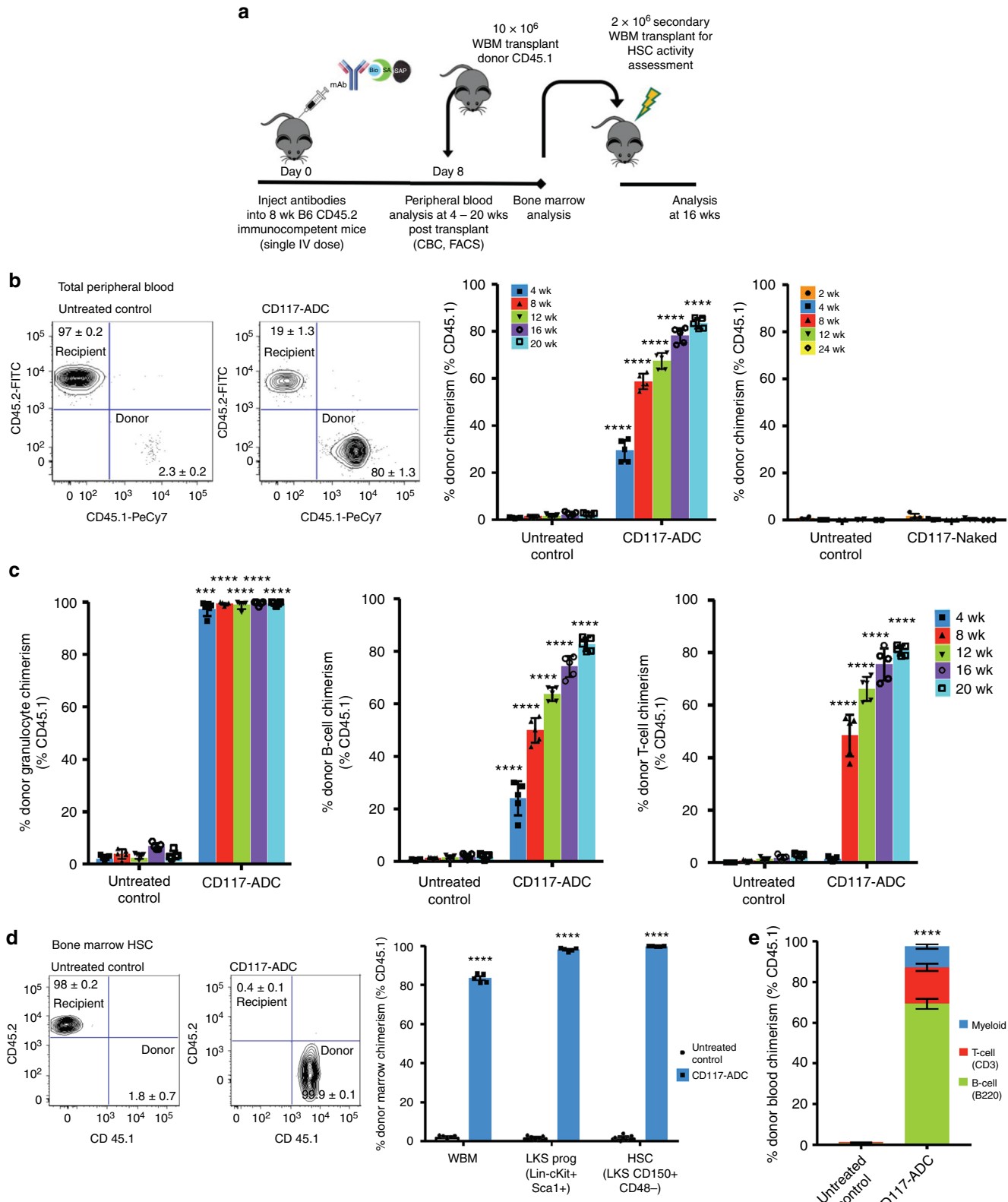

transplantation following CD117-ADC treatment (Fig. 2a). In this setting, optimal engraftment was observed when BM transplantation was performed 8–9 days post CD117-ADC administration (Supplementary Fig 3a, b). Under such conditions as early as 4 weeks after transplantation, we observed >98% donor

myeloid chimerism within the peripheral blood for both donor cell types in mice conditioned with 1.5 mg/kg CD117-ADC but not the naked CD117 antibody, and subsequently over time with further hematopoietic turnover, all peripheral blood compartments became mostly donor-derived by 20 weeks post

**Fig. 2** CD117-ADC conditioning durably and robustly enhances donor murine whole bone marrow (WBM) engraftment. **a** Experimental outline for assessing the ability of antibody-drug-conjugates to condition immunocompetent, wild-type C57BL/6 mice allowing for efficient engraftment of donor murine WBM. **b** CD117-ADC pre-treatment 8 days before infusion of $10 \times 10^6$ CD45.1+ donor whole bone marrow cells leads to robust turnover of recipient peripheral blood as assessed through flow cytometric (FACS) analysis unlike naked CD117 pre-treatment. Representative FACS plots of peripheral blood showing >80% total donor peripheral blood CD45.1+ cells in otherwise CD45.2+ host. **c** CD117-ADC conditioning with WBM transplantation results in rapid multi-lineage engraftment with kinetics paralleling lifespan of cell populations. **d** Almost complete donor HSC (Lin−cKit+Sca+CD150+CD48−) and HPC (Lin−cKit+Sca+) chimerism post CD117-ADC treatment and donor WBM transplantation confirmed by phenotypic bone marrow analysis of transplanted animal 20 weeks post transplantation. Representative FACS plots of bone marrow HSCs showing >99.9% donor HSC chimerism. **e** Donor HSC engraftment confirmed through secondary transplantation of WBM into lethally irradiated recipients. Data represent mean ± SEM ($n = 5$ mice/group assayed individually). Statistics calculated using unpaired $t$ test; all data points significant as indicated vs. untreated control (****$P < 0.0001$)

CD117-ADC conditioning and transplantation (Fig. 2b, c; Supplementary Fig 3c–f, Supplementary Fig 4). Engraftment was found to correlate with CD117-ADC dose with significant donor chimerism already observed with 0.3 mg/kg CD117-ADC dose and maximum engraftment observed with 1.5 mg/kg CD117-ADC dose (Supplementary Fig 3g). To confirm true HSC replacement, recipient mice were sacrificed and BM was assessed to reveal that post 1.5 mg/kg CD117-ADC conditioning and WBM transplantation >99% of HSCs were of donor origin (Fig. 2d). Additionally, when BM from these primary recipients was transplanted into lethally irradiated secondary recipients, high levels of donor-derived multi-lineage engraftment were observed, confirming bona fide stem cell function (Fig. 2e). Total donor chimerism post CD117-ADC-conditioning was similar to that achieved using CD45.2-ADC or conventional reduced-intensity 5 Gy TBI conditioning (Fig. 3a); however, myeloid donor engraftment post CD117-ADC was achieved with faster kinetics as compared to conditioning with CD45.2-ADC or 5 Gy TBI (Fig. 3b). Donor B-cell and T-cell turnover was delayed post CD117-ADC and CD45.2-ADC conditioning as compared with 5 Gy TBI; however ultimately higher donor lymphocyte chimerism was reached in the ADC conditioned groups which continued to increase with time (Fig. 3c, d). Moreover, unlike with irradiation, neither CD117-ADC or CD45.2-ADC conditioning resulted in any neutropenia in the post-transplant period (Fig. 3e). In contrast to CD45.2-ADC and 5 Gy TBI, CD117-ADC conditioning also uniquely avoided lymphopenia in this period likely due to the lack of CD117 expression on the lymphocytes (Fig. 3f, g).

To determine if the efficacy profile of this CD117-ADC was unique to the non-antagonistic 2B8 clone, we tested two additional anti-CD117 clones, reported non-antagonistic 3C11 and antagonistic ACK2. Of the tested clones, 2B8 was most efficiently internalized by EML cells (a CD117+ hematopoietic progenitor line dependent on SCF) (Supplementary Fig 5a), though all antibodies were internalized and cytotoxic to EML cells when conjugated to saporin (Supplementary Fig 5b). All three CD117-ADCs were effective at depleting immunophenotypic and functional HSCs in vivo with no notable gross toxicity (Supplementary Fig 5c–e); however, the 2B8-ADC was the most potent. Not surprisingly, when the three CD117-ADCs were used as conditioning agents prior to congenic CD45.1 WBM transplantation, donor engraftment efficacy roughly correlated with the extent of recipient HSC depletion (Supplementary Fig 5f–i). All CD117-ADCs tested effectively depleted recipient HSCs and enhanced donor HSC engraftment to some degree, implicating the CD117-ADC conditioning concept is broadly applicable and not specific to one agent. However, as 2B8-saporin was the most effective CD117-ADC, it was carried forward for subsequent studies.

Although WBM transplantation has wide clinical utility, donor T-cells contained in such grafts can cause GvHD. T-cell-depleted grafts reduce this complication; however, their engraftment has historically been more challenging with delayed immune reconstitution and increased infectious complications[20]. We tested whether CD117-ADC conditioning could overcome this limitation and enable engraftment of purified HSCs (Fig. 4a), which is also relevant to gene therapy settings where optimally these isolated cells are exclusively transplanted to minimize the number of cells manipulated and thereby amount of vector required. As expected, we observed robust engraftment of purified congenic CD45.1 HSCs with CD117-ADC conditioning, which increased in a donor cell dose-dependent manner (Fig. 4b, c) and led to >85% donor granulocyte chimerism at the highest HSC cell dose at late timepoints. Importantly, we also observed faster and more robust immune reconstitution using purified HSCs in CD117-ADC-conditioned animals than other reduced-intensity conditioning regimens (Fig. 4d, e, Supplementary Fig 6a–c), obviating the historic concern of increased infection susceptibility associated with T-cell-depleted grafts[20]. Stem cell engraftment was further confirmed by assessing BM HSC chimerism that matched long-term peripheral blood granulocyte chimerism (Fig. 4f), and through serial transplantation into lethally irradiated secondary recipients (Fig. 4g). As human HSCs have been hypothesized to utilize similar microenvironments to mouse HSCs, we also tested whether the CD117-ADC enabled the engraftment of human HSCs in mice to give rise to irradiation-free xenografts, which can be used to model human hematopoiesis in vivo. Indeed, when we transplanted 25,000 human CD34+ cord blood cells into CD117-ADC-conditioned adult immuno-compromised NSG recipients, we observed robust total peripheral blood donor human chimerism with multi-lineage engraftment that was similar to irradiation-conditioned controls (Fig. 4h–j).

**CD117-ADC spares mature cells and preserves immunity.** As CD117-ADC, CD45.2-ADC, and 5 Gy TBI yielded equivalent levels of total peripheral blood chimerism post WBM transplantation (Fig. 3a), we assayed the toxicities of each in treated mice that did not undergo transplantation. As CD117 is expressed on erythroid progenitors, previous antagonistic anti-CD117 antibody conditioning approaches were found to cause anemia and require red blood cell transfusion[11,13]. However, interestingly, we observed no significant anemia with either CD117-ADC or CD45.2-ADC treatment (Fig. 5a). Additionally, despite CD117 expression on megakaryocyte progenitors, CD117-ADC-treated recipients experienced only a minor decrease in platelet counts (Fig. 5b). Neither CD117-ADC nor CD45.2-ADC was myeloablative, with both resulting in an increase in myeloid cells (Fig. 5c); however, CD117-ADC uniquely spared lymphoid cells and did not cause profound lymphopenia seen with other methods (Fig. 5d, e).

As viral infections are a significant cause of morbidity and mortality with traditional hematopoietic cell transplantation, treated recipients were challenged with a murine T-cell dependent virus, lymphocytic choriomeningitis virus (LCMV), and assessed for the ability to generate viral-specific responses. Only post CD117-ADC treatment were LCMV-specific T cell

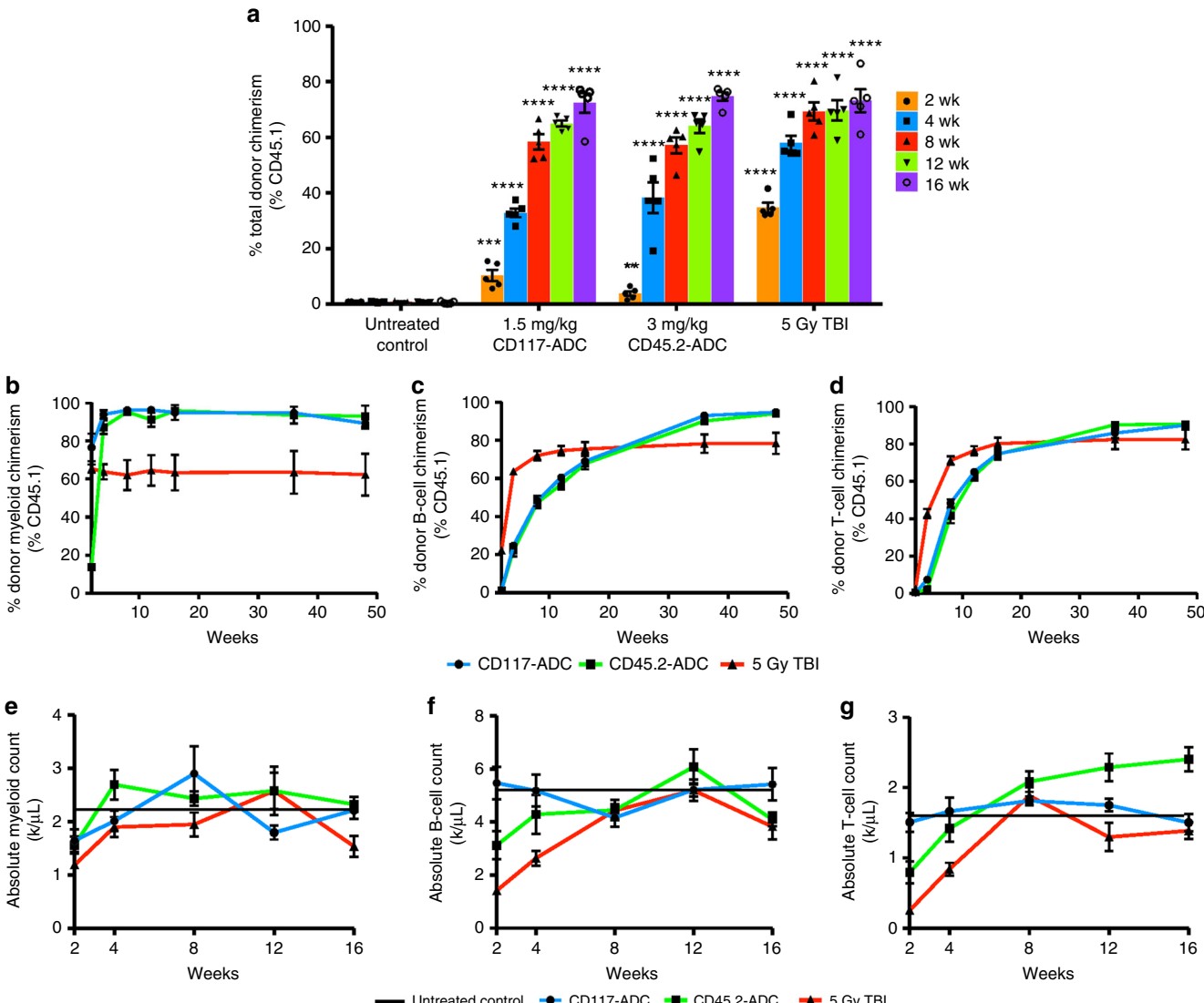

**Fig. 3** CD117-ADC conditioning leads to comparable total multi-lineage engraftment as alternative conditioning regimens post WBM transplantation with improved engraftment kinetics. **a** CD117-ADC conditioning pre-WBM transplantation leads to comparable total peripheral blood chimerism as CD45.2-ADC and 5 Gy TBI conditioning. Robust multi-lineage donor engraftment with rapid donor granulocyte chimerism (**b**) and slower donor B-cell (**c**) and T-cell turnover (**d**). **e**–**g** No significant leukopenia observed post CD117-ADC treatment and WBM transplantation, as opposed to CD45.2-ADC and 5 Gy TBI conditioning with grossly normal absolute myeloid cells (**e**), absolute B-cells (**f**), and absolute T-cells (**g**). Data represent mean ± SEM ($n = 5$ mice/group, assayed individually). Statistics calculated using unpaired $t$ test; all data points significant as indicated vs. untreated control (**$P < 0.01$; ***$P < 0.001$; ****$P < 0.0001$)

responses robustly observed (Fig. 5f). Additionally, viral-specific immunity was preserved after this treatment, and many LCMV-specific T cells were observed in recipients that had previously been challenged with LCMV and then treated with CD117-ADC (Fig. 5g). Fungal immunity was also preserved following CD117-ADC treatment, as after intravenous challenge with the human fungal pathogen, *Candida albicans*, CD117-ADC recipients could mount effective neutrophil responses as evidenced by decreased kidney fungal organ loads with increased survival (Fig. 5h). In contrast, busulfan and irradiation conditioning resulted in profound neutropenia, high *C. albicans* organ loads, and early mortality (Fig. 5h).

Even at the highest 1.5 mg/kg CD117-ADC dose, this treatment was well-tolerated and non-myeloablative; moreover, treatment permitted long-term survival (>6 months) without requiring hematopoietic cell transplantation or transfusions unlike alternative treatments such as irradiation

conditioning or ACK2 with CD47 blockade[13]. As CD117 is also expressed on a variety of other rare cell types throughout the body such as melanocytes and germ cells, toxicity assessment including full necropsy was performed after CD117-ADC treatment. Unlike post alternative conditioning strategies such as TBI or ACK2, recipients appeared healthy with full coat color and remained fertile with vigor (Supplementary Fig 7a). At the highest 1.5 mg/kg optimal treatment dose, CD117-ADC conditioning did result in mild liver toxicity with transiently elevated transaminases and rare apoptotic hepatocytes (Supplementary Fig 7b), and at increased doses animal mortality was intermittently observed. However, transaminitis resolved without intervention and no other toxicity was documented at this optimal dosing with subsequent resolution of all histologic liver changes. Upon review by an experienced pathologist, all other organs examined had normal histology post treatment (Supplementary Fig 7c). Collectively, these results indicate that

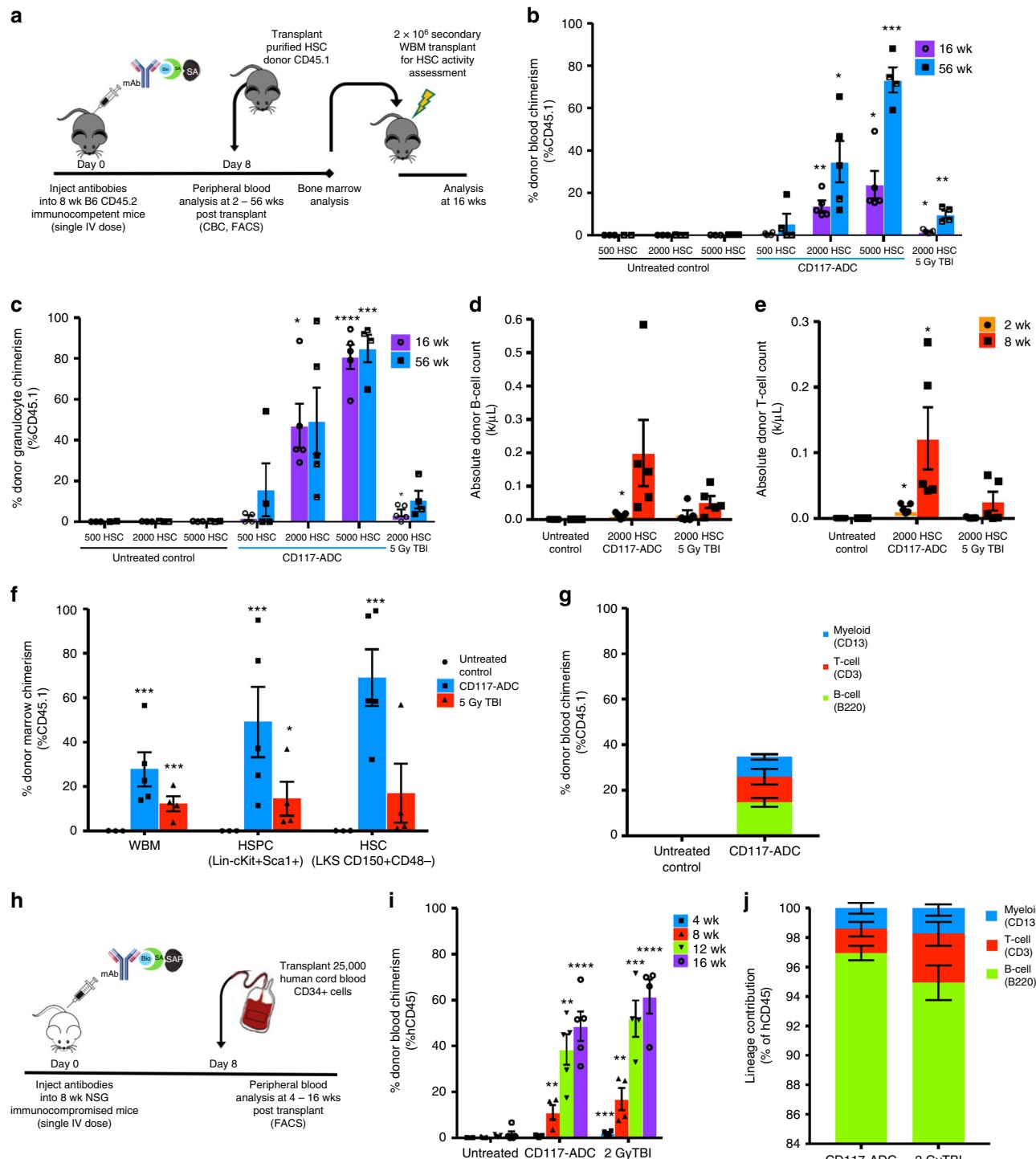

**Fig. 4** CD117-ADC conditioning effectively enhances purified mouse and human HSC engraftment, with increased HSC cell dose resulting in increased multi-lineage reconstitution. **a** Experimental outline for assessing the ability of antibody-drug-conjugates to condition immunocompetent, wild-type C57BL/6 mice allowing for efficient engraftment of donor murine HSC. **b**, **c** CD117-ADC pre-treatment 8 days before infusion of FACS-purified CD45.1+ donor HSCs leads to robust total donor peripheral blood (**b**) and donor granulocyte chimerism (**c**). Rapid donor B-cell (**d**) and T-cell (**e**) reconstitution post transplantation of purified HSCs into CD117-ADC-conditioned animals as compared to 5 Gy TBI controls. **f** Similarly enhanced donor HSPC (Lin−cKit+Sca+) and HSC (Lin−cKit+Sca+CD150+CD48−) chimerism post CD117-ADC treatment and transplantation of 2000 purified HSCs confirmed by phenotypic bone marrow analysis of transplanted animal 20 weeks post transplantation. **g** Donor HSC engraftment confirmed through secondary transplantation of WBM into lethally irradiated recipients. **h** Experimental outline for assessing the ability of antibody-drug-conjugates to condition immunocompromised, NSG mice for efficient engraftment of donor human CD34+ cord blood cells. **i** Single treatment of CD117-ADC effectively enhances donor human cord blood HSPC engraftment in NSG mice, enabling creation of irradiation-free xenografts with multi-lineage reconstitution with total donor chimerism nearing similar levels as 2 Gy TBI. **j** Multi-lineage human chimerism observed in all xenografted mice, with B-cell engraftment predominant regardless of the conditioning method. Data represent mean ± SEM ($n = 3$–5 mice/group, assayed individually). Statistics calculated using unpaired $t$ test; all data points significant as indicated vs. untreated control (*$P < 0.05$; **$P < 0.01$; ***$P < 0.001$; ****$P < 0.0001$)

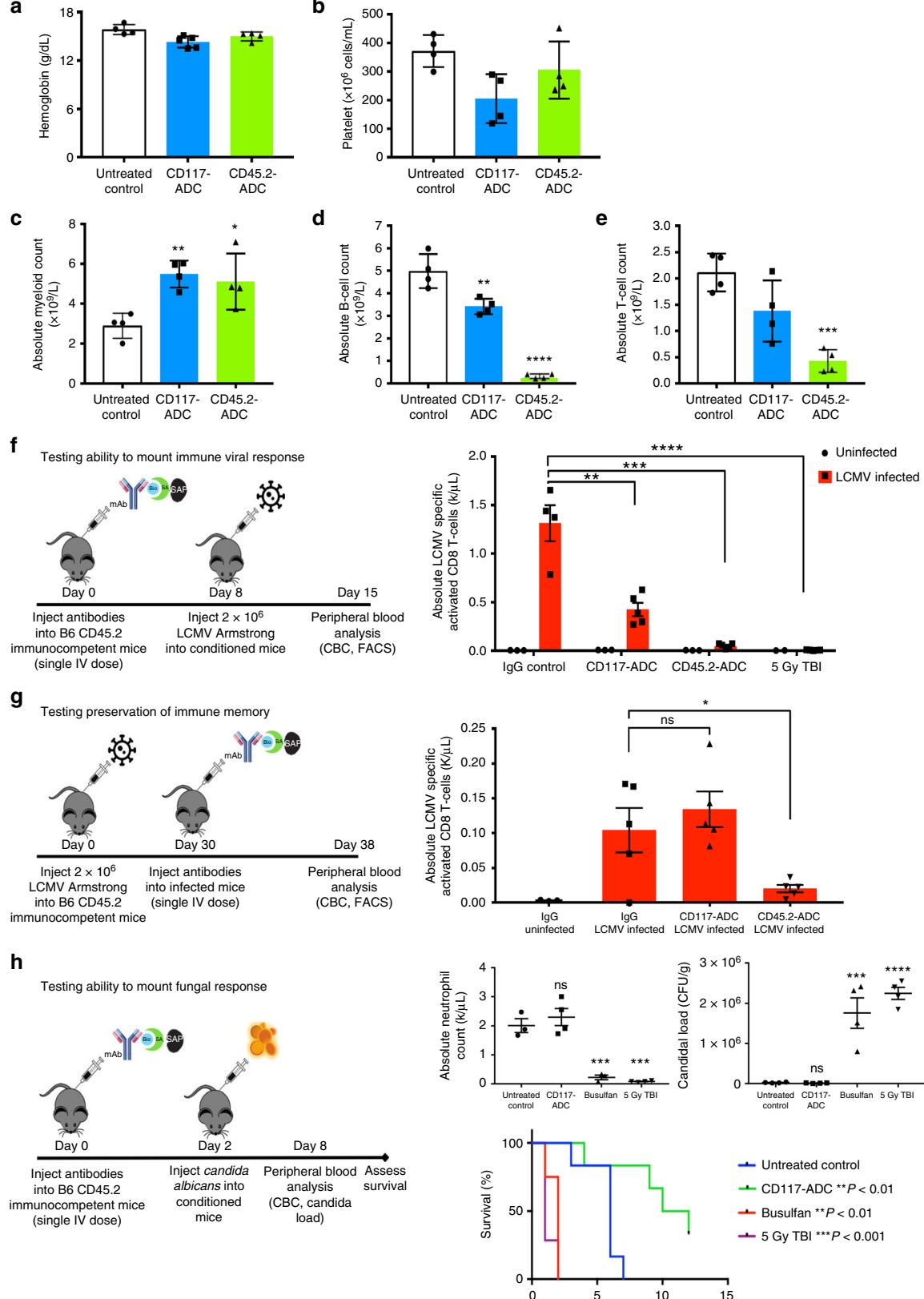

CD117-ADC is less detrimental systemically than conventional conditioning, while achieving efficient HSC depletion enabling rapid, high, sustained donor BM/HSC engraftment and preservation of immunity allowing for protection post viral and fungal challenge. Furthermore, we found that alternative HSC antigens, including CD27, CD90, CD110, and CD184 even in non-optimized settings, could be similarly targeted to generate further restricted conditioning agents implying broad applicability of such an HSC-ADC approach (Supplementary Fig 8a, b).

**Fig. 5** CD117-ADC is uniquely non-ablative to the peripheral blood with no clinically significant cytopenias, and results in preservation of phenotypic and functional immunity. **a**, **b** Despite HSC depletion, only very minor and not clinically significant decreases in hemoglobin (**a**) and platelet counts (**b**) were observed 8 days post CD117-ADC treatment. **c** Unlike TBI, ADC treatments did not lead to neutropenia, and myeloid counts were increased 8 days post CD117-ADC treatment at the time of transplantation. Unlike CD45.2-ADC, peripheral B-cell (**d**) and T-cell counts (**e**) remained largely intact 8 days post administration of CD117-ADC. **f** T-cell numbers not only remained intact post CD117-ADC treatment, but animals were uniquely able to mount viral immune responses to LCMV infection post CD117-ADC as indicated in the experimental outline with results showing present LCMV-specific activated CD8 T-cells. **g** Additionally, post CD117-ADC treatment LCMV-specific activated CD8 T-cells generated from prior LCMV infection uniquely remained, as indicated in the experimental outline with results showing similar numbers of LCMV-specific activated CD8 T-cells CD117-ADC similar to post no conditioning. **h** Animals were also able to mount functional immune response to *Candida* challenge post CD117-ADC treatment, as indicated in the experimental outline with animals showing persistent neutrophils, control of *Candida* load, and increased survival compared to other treatment groups. Data represent mean ± SEM ($n = 3–5$ mice/group, assayed individually). Statistics calculated using unpaired $t$ test; all data points significant as indicated vs. untreated control (*$P < 0.05$; **$P < 0.01$; ***$P < 0.001$; ****$P < 0.0001$)

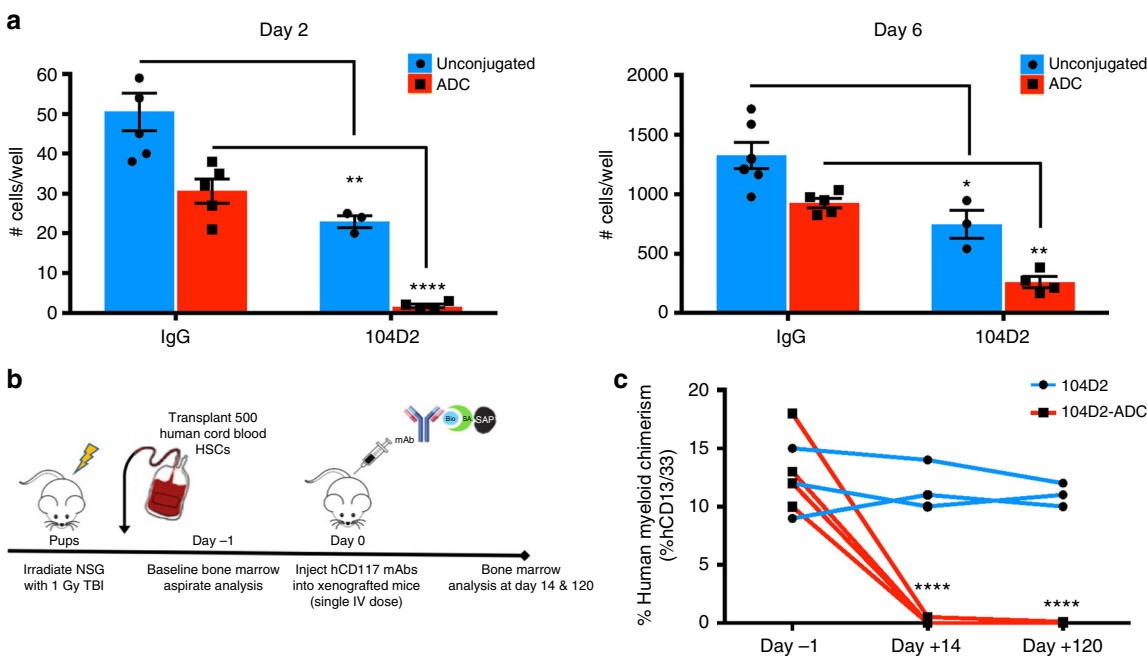

**Fig. 6** Human CD117-ADC inhibits human HSCs in vitro and robustly depletes human HSCs in vivo. Human CD117-ADCs were prepared similarly by conjugating biotinylated anti-human CD117 (104D2 clone) with streptavidin–saporin toxin. **a** Human CD117-ADC inhibited human cord blood growth in vitro, as compared to unconjugated antibody controls. **b** Experimental outline for assessing the ability of anti-CD117 antibodies and antibody-drug-conjugates to deplete human HSCs in xenograft mice. **c** Only antibody conjugated to saporin is effective at eliminating human myeloid cells. Data represent mean ± SEM ($n = 3–5$ samples/group, assayed individually). Statistics calculated using unpaired $t$ test; all data points significant as indicated vs. unconjugated antibody-treated control animals (*$P < 0.05$; **$P < 0.01$; ****$P < 0.0001$)

**CD117-ADC depletes human HSCs.** Although the CD117-ADC generated with the 2B8 clone enabled powerful, proof-of-concept studies and serves as an important tool, this agent only targets mouse CD117. We tested whether similar CD117-ADCs could potently deplete human HSC by first creating a human CD117-ADC using the biotinylated anti-human CD117 (clone 104D2) with streptavidin–saporin linkage. Parallel studies showed that this human CD117-ADC inhibited human HSC growth in vitro (Fig. 6a). This human CD117-ADC was then administered to humanized xenograft mice, and in vivo human HSC depletion was assayed by measuring human myeloid cells in the peripheral blood which are human HSC and progenitor derived (Fig. 6b). Human HSC activity as represented by the inability to produce human myeloid cells was found to be diminished 440-fold by this human CD117-ADC (Fig. 6c). These results imply that similar human CD117-ADCs can deplete human HSCs, suggestive of potential clinical application of such a strategy in conditioning prior to hematopoietic cell transplantation in patients. Given these results, we are optimistic of the rapid advancement and development of similar clinical-grade CD117-ADCs.

## Discussion

Historically, non-selective chemotherapy, either alone or with irradiation, has been necessary for effective hematopoietic cell transplantation[10]. These classical cytotoxic interventions drive the morbidity and mortality of HSCT and lead to many of the complexities of the procedure[7]. Even at reduced doses, which are described as "non-myeloablative", some myeloid ablation with non-specific organ toxicity occurs[9]. We previously demonstrated that host HSC depletion is a critical requirement to enabling donor HSC engraftment and long-term hematolymphoid turnover, and can be achieved using specific HSC-targeting strategies[11]. This study uniquely highlights that long-term donor BM or HSC engraftment can be achieved in a setting with exclusive HSC depletion. Moreover, donor engraftment occurs with mostly intact host hematopoietic progenitor and myeloid cells, highlighting successful transplantation in a true non-marrow and non-myeloablative context. These studies question the common synonymous use of the term "myeloablation" with "HSC ablation", as here we display strong evidence that the two can be separated. Based upon this work, we further advocate for the

development of antibody-based conditioning strategies that selectively deplete host HSCs. We primarily targeted CD117 in these studies but given our results, presumably any HSC-specific antigen could be targeted, though the strategy we employed herein also likely requires antigen/antibody internalization to effectively deliver the drug.

Antibody-based methods targeting HSCs are advancing to patients, with the most advanced antagonistic anti-CD117 naked antibody derived from our prior work already in clinical trials in severe combined immunodeficiency patients which is showing encouraging clinical results[21]. However, although such approaches are very promising for certain patient groups, all prior reported methods have significant limitations. To address these challenges, we show here that a single-agent CD117-ADC is a superior conditioning agent which safely and effectively enables robust BM/HSC transplantation in an immunocompetent setting while uniquely preserving immunity. This approach enables transplantation while avoiding clinically significant collateral damage across tissues. In particular, no anemia, neutropenia, lymphopenia, coat color discoloration, or infertility was observed. We foresee that the preservation of immunity would minimize additional severe infectious co-morbidities associated with conventional conditioning, making HSCT safe and potentially achievable as a simple outpatient procedure. Furthermore, the preservation of past immunity makes this approach additionally uniquely attractive. These features would be especially useful in settings of autologous gene therapy or gene editing where immune depletion is not required and presents no added benefit as the host generally does not reject its own gene-modified cells. As current conditioning methods are a major limitation for gene therapy transplantation, successful translation of our approach to humans could dramatically enhance the utility of gene therapy and gene editing across disease areas ranging from sickle cell anemia, beta thalassemia, congenital immunodeficiencies to HIV and potentially even broadly enable HSCs to be used as a vector for any type of protein production.

Although transient immune suppression would be necessary in the allogeneic setting to prevent immune-mediated donor graft rejection, we predict that CD117-ADC conditioning may still have broad adoption in this setting with advantages over classic conditioning. In addition to decreased toxicity and decreased tissue inflammation, CD117-ADC conditioning enables efficient transplantation of purified, T-cell-depleted grafts which likely would result in significantly decreased GvHD[20]. This approach could also enable more rapid lymphoid recovery which is a major concern with manipulated grafts and further enable shorter immune suppression with decreased infectious complications. Additionally, such CD117-ADC conditioning may lead to deeper remissions in malignant settings, especially for patients with leukemias expressing CD117 and/or when combined with conventional therapies. Moreover, in complementary studies with combined CD117-ADC and transient immune suppression, such conditioning enables unique fully HLA-mismatched transplantation and concurrent immune tolerance likely due to robust donor HSC engraftment (Li et al., co-published) enabling transplantation with many more diverse donors.

Our data suggest that a single-agent CD117-ADC may be the conditioning agent of choice for both mouse and human. Although the CD117-ADCs to saporin through biotin–streptavidin linkage developed here are proof-of-concept agents, with proper development and optimization, equivalent anti-human CD117-ADCs may be extremely powerful and dramatically expand the application of HSCT. As saporin toxin is known to have some hepatotoxicity[22], which we similarly observed in our studies, alternative drug-conjugates may be preferred and utilized in similar fashions to further improve the

safety profile of such a conditioning approach[23]. However, saporin may also be a viable ADC, given that it has been previously effectively used in clinical trials and in this type of conditioning setting would only need to be used as a one-time administration. Additionally, although some transient hepatotoxicity was seen with this CD117-ADC at the maximum 1.5 mg/kg dose, this is still likely a much-improved toxicity profile over current standard-of-care chemotherapy and/or irradiation conditioning. Furthermore, we have shown that a fivefold decreased dosing of this CD117-ADC can also generate therapeutically meaningful levels of low mixed chimerism. Such dosing may be ultimately preferred for many diseases which would be expected to have minimal or negligent hepatotoxicity. Interestingly, although CD117 is expressed on other rare cells throughout the body, such as germ cells and melanocytes, no toxicity was documented in these tissues after this CD117-ADC administration unlike after alternative conditioning treatments. Moreover, post this treatment, many CD117-expressing hematopoietic progenitors were also surprisingly spared in the BM despite robust HSC depletion, further implicating that not all CD117-expressing cells are killed by this CD117-ADC treatment, which likely further contributes to its improved toxicity profile.

We anticipate such agents to not only be effective at enabling robust HSCT, but also be gentler to other tissues and the hematopoietic microenvironment itself especially with further optimization. As such, this may permit revisiting basic aspects of hematopoiesis since much of our scientific understanding of both human and murine hematopoiesis is based upon transplantation into irradiation or chemotherapy-conditioned hosts. The significant tissue damage associated with classic methods may have introduced artifacts which potentially confounded interpretations of prior studies. Therefore, in addition to its clinical potential for treatment of patients, CD117-ADC conditioning in experimental transplantation may also be a useful tool that provides new scientific insights. Given these findings, we anticipate CD117-ADC conditioning to be broadly used in both experimental and clinical transplantation, possibly expanding the use of this powerful curative procedure across disease settings.

## Methods

**Mice**. About 8–10-week-old female C57BL/6J (CD45.2+) mice were purchased from Charles River or Jackson Laboratories and 7-week-old female NOD.Cg-Prkdc[scid] Il2rg[tm1Wjl]/SzJ mice were purchased from Jackson Laboratory and used as recipients in all studies. Newborn NOD.Cg-Prkdc[scid] Il2rg[tm1Wjl]/SzJ pups were used to generate xenografts for human CD117-ADC studies, and recipients were used 14 weeks post xenograft generation. About 10–12-week-old female B6.SJL-Ptprc[a]Pepc[b]/BoyJ (B6.SJL, CD45.1+) mice were purchased from Jackson Laboratories and used as donors for most C57BL/6J (CD45.2+) mice experiments. In additional experiments, 10–12-week-old female B6 GFP+ (GFP+) mice were purchased from Jackson Laboratories and used as donors. All animal experiments were performed in compliance with the Institutional Animal Care and Use Committee and approved by Harvard Medical Area Standing Committee or Stanford University Administrative Panel on Laboratory Animal Care.

**Antibody-drug-conjugate generation and administration**. Biotinylated anti-CD117 (Biolegend; clones 2B8 [Lot # B170777, B205356], 3C11, ACK2, 104D2), biotinylated anti-CD45.2 (Biolegend; clone 104), biotinylated anti-CD27 (Biolegend; clone LG3A10), biotinylated anti-CD90.2 (Biolegend; clone 30-H12), biotinylated anti-CD110 (Clontech; clone AMM2), and biotinylated anti-CD184 (Biolegend; clone L276F12) were combined with streptavidin-SAP conjugate (Lot # 94-31; Advanced Targeting Systems) in 1:1 molar ratio and then diluted in PBS to desired concentration. CD117-ADCs were generally administered at 1.5 mg/kg (~12 μg of streptavidin-saporin) per recipient, unless otherwise indicated. CD45.2-ADCs were generally administered at 3 mg/kg (~24 μg of streptavidin-saporin) per recipient, as previously optimized. Additional HSC antigen ADCs were administered as follows: CD27-ADCs at 1.5 mg/kg, CD90-ADCs at 3 mg/kg, CD110-ADCs at 3 mg/kg, and CD184-ADCs at 3 mg/kg. Alternative reagent lots were utilized in subsequent studies with slightly altered properties requiring further optimization. Antibody-drug-conjugates were administered in 300 μL via retro-orbital intravenous injection. Naked anti-CD117 antibody (BioXCell; clone 2B8) was administered in separate experiments at 500 μg (~25 mg/kg) per recipient as a control.

**Murine BM and HSC transplantation**. BM cells were harvested from donor CD45.1 or GFP mice. WBM cell numbers were determined by counting via CBC (complete blood count) using VetScan HM5 (Abaxis); $10 \times 10^6$ WBM cells or 500–5000 purified HSCs were transplanted via retro-orbital intravenous infusion 8 days post antibody-drug-conjugate injection or 1 day post 5 Gy TBI into C57BL/6J (CD45.2+) immunocompetent mice as previously described[16]. Purified mouse HSCs used in transplantation studies were obtained by double sorting via fluorescence-activated cell sorting (FACS) Lin−cKIT+Sca1+CD150+CD34− cells on a BD FACS Aria II, as previously described[11]. Competitive transplants were performed by transplanting $1 \times 10^6$ WBM cells from treated C57BL/6J (CD45.2+) animals and $1 \times 10^6$ WBM cells from untreated B6.SJL (CD45.1+) animals into 9 Gy, lethally irradiated, B6.SJL (CD45.1+) wild-type recipients as previously described[16]. Secondary transplants were performed by transplanting $2 \times 10^6$ WBM cells from transplanted animals into 9 Gy, lethally irradiated, C57BL/6J (CD45.2+) wild-type recipients as previously described[16].

**Human cord blood transplantation**. Single donor-derived cord blood CD34+ cells were obtained from AllCells (Alameda, CA) under local IRB-approved protocol with written informed consent. Cells were counted via hemocytometer and suspended to desired dilution of $2.5 \times 10^4$ cells/200 μL. Each animal received $2.5 \times 10^4$ cells via retro-orbital intravenous infusion 8 days post antibody-drug-conjugate injection or 1 day post 2 Gy TBI into adult NSG immunodeficient mice. To test for in vivo efficacy of human CD117-ADCs, xenografts were generated by transplanting 500 FACS-sorted human HSCs (Lin−CD34+CD38−CD90+) via facial vein injection into newborn NSG immunodeficient mice post 1 Gy TBI as per previously described methods[24].

**Peripheral blood analysis**. Peripheral blood was sampled from treated animals at 8 days post antibody-drug-conjugate administration to assess for peripheral blood effects. Transplanted mice were assessed via peripheral tail vein bleeds at 2, 4, 8, 12, 16 weeks or later as indicated post donor cell transplantation. White blood cells, hemoglobin, platelets, absolute lymphocyte count, and absolute neutrophil count were determined by a VetScan HM5 (Abaxis) or a Hemavet 950 (Drew Scientific). In addition, the blood samples were red blood cell-lysed and fixed using BD Staining buffer, and subsequently stained for CD3 (T-cells), B220 (B-cells), Gr1 (Granulocytes), CD45.1 (donor), and CD45.2 (host) to assess chimerism by flow cytometry. Xenografts were assessed via peripheral tail vein bleeds at indicated timepoints. Blood samples were similarly red blood cell-lysed and stained for hCD45 (pan-blood), hCD13 (myeloid), hCD3 (T-cell), and hCD19 (B-cell) to assess human chimerism by flow cytometry. Data analysis was done on FlowJo 10 software (version 10.2; Treestar, Ashland, OR) using previously published methods[11].

**BM analysis**. BM cells were harvested post sacrifice and single cell suspensions were obtained by crushing lower extremity bones. Cells were filtered and stained for appropriate markers to quantify HSC and progenitors and to determine donor chimerism. Samples were run on flow cytometry using various panels to assess cellular makeup including: CD45.1, CD45.2, CD117, CD34, CD48, CD41, Flk2, CD105, CD150, Sca1, FcGammaRa, and IL7Ra (all purchased from Biolegend). Data analysis was done on FlowJo 10 software (version 10.2; Treestar, Ashland, OR) using previously published subset definitions[25]. Colony forming cell (CFC) assays were performed by plating 25,000/mL WBM cells in M3434 methyl cellulose media (Stem Cell Technologies) in 3 cm dishes. After 7 days of incubation, the total number of colonies formed were counted using a light microscope.

**In vitro cell death assays**. In vitro cell death experiments were performed using the EML (ATCC CRL-11691) cell line cultured in IMDM media in the presence of 200 ng/mL murine stem cell factor (mSCF1, R&D Systems). Cells were plated in 96-well plates with 5000 cells/well in 100 μL volume cell culture media containing various concentrations of antibody-drug-conjugate. Three independent experiments were performed with three technical replicates within each experiment. After 72 h, cell viability was determined using the CellTiter MTS assay (Promega). PBS-treated and 10 μM staurosporin-treated cells (Sigma) were used as live and dead controls, respectively. Additionally, in vitro human HSC cell death experiments were performed using FACS-sorted primary human cord blood HSCs. Twenty-five human cord blood HSCs (Lin−CD34+CD38−CD90+CD45RA−) were sorted using previously described methods[25] into 96-well plates with StemSpan media supplemented with 100 ng/mL of human cytokines SCF, TPO, FLT3, and IL3 (R&D Systems) as well as 10 μg/mL of biotinylated antibodies or saporin conjugates. Cells were quantified by counting under light microscopy 2 days and 6 days post plating.

**Measurement of antibody internalization**. EML cells (200,000/mL) were plated into 96-well plates in complete media (IMDM, 10% FBS, 200 ng/mL mSCF) and 20 nM of a 1:1 mixture of biotinylated antibodies with streptavidin-AF488 (Life Technologies) with $n = 6$ technical replicates. After 4 h of incubation, the cells were washed twice and resuspended in PBS containing 2% FBS. Samples were split into two and one sample was incubated with 0.25 mg/mL polyclonal anti-AF488 quenching antibody (clone A-11094, Life Technologies). AF488 signal in samples with and without quenching antibody was quantitated by flow cytometry. Unstained and time zero stained controls (stained on ice) were used to determine the quenching efficiency on the AF488-quenching antibody and calculate internalization frequency of the test samples.

**Toxicity assessment**. C57BL/6 mice were sacrificed at various timepoints (2 days or 1 year after treatment), submerged individually in 300 mL containers of Bouin's solution (Sigma) or 10% Formalin solution (VWR) and delivered to the Harvard Medical Area Core Specialized Histopathology Services or Stanford Department of Comparative Medicine's Animal Histology Services. Necropsy with resulting H&E histology was performed using standard methods. Pathology was assessed by a trained and certified veterinary or hematopathologist. Liver function testing was also assessed in additional C57BL/6 mice (2 or 4 days after treatment) via Charles River Clinical Pathology Services.

**Immune memory and response with LCMV**. LCMV Armstrong was a gift from Dr. John Wherry (University of Pennsylvania, Philadelphia, PA) and was propagated according to the standard protocol[26]. LCMV ($2 \times 10^5$ pfu) was intraperitoneally injected into 6–12-week-old C57BL/6 mice. LCMV-specific CD8 T-cells were stained by PE-conjugated H2DbGP33-KAVYNFATM tetramers, which was kindly provided by the NIH Tetramer Facility (Atlanta, GA). Irradiation post LCMV infection was not possible due to facility policies.

**Systemic challenge with *C. albicans***. *C. albicans*, wild-type stain SC5314 (ATCC MYA-2876), was grown overnight from frozen stocks in yeast extract, peptone, and dextrose (YPD) medium (BD Biosciences) with 100 μg/mL ampicillin (Sigma) in an orbital shaker at 30 °C. After pelleting and washing with cold PBS, yeasts were counted using a hemocytometer and cell density adjusted in PBS to 150,000 yeasts per 200 μL. C57BL/6 mice (non-conditioned control, 8 days post 1.5 mg/kg intravenous CD117-ADC, 4 days post 20 mg/kg intraperitoneal busulfan, or 2 days post 5 Gy TBI) were injected via lateral tail vein with 150,000 yeasts, and the animals were monitored daily. Moribund mice were euthanized humanely. For kidney *Candida* organ loads, mice were sacrificed, kidneys harvested, and homogenized using a hand-held tissue homogenizer for 30 s. Organ homogenates were diluted in PBS and plated on YPD solid agar. *Candida* colonies were counted at 48 h. Fungal burden was expressed as CFU per gram of kidney.

**Statistics**. Animal groups were typically $n$ of five for all experiments, and experiments were performed in duplicate unless otherwise noted. All statistics were calculated using unpaired $t$ tests using two-sided analysis except Kaplan–Meier data, which were analyzed by log-rank (Mantel–Cox) test. Alphanumeric coding was used to blind pathology samples and CFC counting.

**Reporting Summary**. Further information on experimental design is available in the Nature Research Reporting Summary linked to this article.

## Data availability
The authors declare that all the data supporting the finding of this study are available with the paper and its supplementary information files.

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

## Acknowledgements

The authors thank M. Bamberg and C. Klein for laboratory management; A. Zguro for animal care; A. Rybak for technical assistance; R. T. Bronson and K. Casey for histological expertise; N. Barteneva and K. Ketman and the Harvard Stem Cell and Regenerative Biology flow cytometry core for flow cytometry assistance; and R. Perriman, P. Murphy, and M. Cooke for critical reading of the manuscript. A.C. was supported by a Potter Fellowship to the Boston Children's Hospital Trust. R.P. was supported by a Life Science Research Foundation fellowship sponsored by the Jake Wetchler Foundation and grants from the Harvard Blavatnik Biomedical Accelerator Fund. B.S. was supported by an American Society of Hematology scholar award, ISCIII PI17/01346 and AEFAT. J.H. was supported by an NIH NHLBI K99/R00 HL119559. W.W.P. supported by a Walter V. and Idun Berry Postdoctoral Fellowship. M.K.M. was supported, in part, by NIAID NIH R01 AI 132638. J.A.S. was supported by the California Institute for Regenerative Medicine (DR2A-05365), the Gunn/Olivier Research Fund, the Virginia and D.K. Ludwig Fund for Cancer Research, the Stinehart-Reed Foundation, and the HL Snyder Medical Foundation. D.T.S. was supported by the Gerald and Darlene Jordan Chair of Medicine of Harvard University and grants from the Harvard Blavatnik Biomedical Accelerator Fund, NIH NHLBI HL44851 (D.T.S.), and HL129903 (D.T.S., D.J.R.). D.J.R. was supported by grants from the NIH (RO1HL107630, R00AG029760, and UO1DK072473-01) as well as grants from The Leona M. and Harry B. Helmsley Charitable Trust, The New York Stem Cell Foundation, The Harvard Stem Cell Institute, and the American Federation for Aging Research.

## Author contributions

A.C., R.P., A.S., Y.H., and M.K.M. designed the research, performed the experiments, interpreted the data, and wrote the article. J.H., B.S., W.W.P., T.A.T., Y.Y.C., E.W., and G.W. designed the research, performed the experiments, and interpreted the data. J.A.S. designed the research and interpreted the data. F.W., D.T.S., and D.J.R. designed the research, interpreted the data, and wrote the article.
