## [Peer Review File · Nature Communications]

Reviewers' comments:

Reviewer #1 (Remarks to the Author):

The authors have not performed a single experiment to answer my comments. The problem of small numbers of mice per group was not addressed. The hepatotoxicity was termed "no ideal" without any attempt to address this concern in more detail.

Important controls are missing as also stated by the authors: "This was not tested as unfortunately we could not find an animal facility for which we could perform irradiation in virally infected animals."

In my view this manuscript does not meet the quality criteria for Nature Communications.

Reviewer #2 (Remarks to the Author):

General comments:

This manuscript by Czechowicz et al. describes the HSC ablation using CD117-antibody-drug-conjugates enabling effective HSC transplantation with preservation of immunity. This is an interesting paper with a clear translational perspective, providing new options in conditioning during HSC transplantation.

Major points:

- The authors state that CD117-Saporin "specifically depletes host HSCs". As CD117 (c-kit) is also expressed on other stem and progenitor cells, the authors should consider rewording this. Additionally, this point should be explicitly mentioned and discussed, e.g. c-kit expression in germ stem cells and melanocytes. It is encouraging that the authors write "As CD117 is also expressed on a variety of other rare cell types throughout the body, full toxicity assessment was performed after CD117-ADC treatment." However, this reviewer feels that some more information on this topic should be given, e.g. in the discussion.
- In the view of this reviewer the comments by reviewers 1 and 3 have been adequately addressed.

REVIEWER #1:

Remarks to the Author: *The authors have not performed a single experiment to answer my comments. The problem of small numbers of mice per group was not addressed. The hepatotoxicity was termed "no ideal" without any attempt to address this concern in more detail. Important controls are missing as also stated by the authors: "This was not tested as unfortunately we could not find an animal facility for which we could perform irradiation in virally infected animals." In my view this manuscript does not meet the quality criteria for Nature Communications.*

Response: We respectfully disagree with this reviewer and would argue that we have addressed the reviewer's prior concerns by appropriately adjusting the text, displaying the data in clearer ways, and have included additional experimental results as requested. Each of the reviewer's past concerns was explained in the prior point-by-point responses that were previously provided to Nature Medicine, although we are not certain this was re-provided to the reviewer in this Nature Communication submission. Included in this was the explanation that in total we have performed these types of transplants in repeated experiments with > 30 animals and are confident of our findings. Each experiment was only conducted with 4-5 mice/group due to the high costs of the reagents which prohibited larger cohorts, however all experiments were performed at least in duplicate. Additionally, due to the very large differences observed between experimental groups, we did not feel it was critical to increase the cohort sizes. We adjusted how we display this data by including data points of individual animals to better highlight the consistency of these results and have also now included statistical significance for each experiment to further illustrate these clear differences (in many cases $p < 0.0001$). Finally, we generated and included additional data to address the reviewers concerns and specifically added Supplemental Figure 4, which includes repeat experiments and additional controls. Additionally, we also conducted additional candida challenge experiments as requested by this reviewer to specifically quantify the effect on neutrophils and fungal load which were included in Figure 5h. In regards, to the mild transient hepatotoxicity observed with the proof-of-concept CD117-ADC used in these studies, we explained and commented on this in detail in the discussion per the review's request specifically stating "As saporin toxin is known to have some hepatotoxicity which we similarly observed in our studies, alternative drug-conjugates may be preferred and utilized in similar fashions to further improve the safety profile of such a conditioning approach. However, saporin may also be a viable ADC given it has been previously effectively used in clinical trials and in this type of conditioning setting would only need to be used as a one-time administration. Additionally, although some transient hepatotoxicity was seen with this CD117-ADC at the maximum 1.5mg/kg dose this is still likely a much-improved toxicity profile over current standard-of-care chemotherapy and/or irradiation conditioning." Furthermore, we included additional studies showing efficacy of the CD117-ADC at reduced doses in Supplemental Figure 2g which should further have decreased hepatotoxicity, which enables an additional opportunity for non-toxic clinical translation of this agent. Given the known hepatotoxicity of saporin, we believe additional experiments investigating hepatotoxicity in this system are outside the scope of this study, especially as the CD117-ADC utilized in these studies is a proof-of-concept agent and may not be the ultimate clinical therapeutic used in patients (as other drug-conjugates are now additionally being explored). Furthermore, we believe all appropriate controls were included in the manuscript and specifically we also included naked CD117 mAb in Figures 1c and Figure 1d at the reviewer's request. We believe additional experiments would take considerable time given the long-nature of

transplantation experiments, and we believe this important work should be shared expeditiously especially given its potential for clinical transplantation and its use as a tool to better study hematopoiesis. We believe this manuscript is of high-impact to both the scientific and clinical community and already have dozens of investigators that have contacted us to obtain and utilize these agents. Thus, we would appreciate this manuscripts publication as is without further experimentation apart from that which we already included.

REVIEWER #2:

General comments: *This manuscript by Czechowicz et al. describes the HSC ablation using CD117-antibody-drug-conjugates enabling effective HSC transplantation with preservation of immunity. This is an interesting paper with a clear translational perspective, providing new options in conditioning during HSC transplantation.*

Response: We appreciate the reviewer's assessment of our work and the noted translatability of our approach. Given our findings, equivalent clinical agents are already in advanced clinical development (Hartigan, et al ASH 2017 and ASBMT 2018 Abstracts), and we are hopeful these will be quite meaningful therapeutics for patients and expand/enable improved hematopoietic cell transplantation.

Major point (#1): *The authors state that CD117-Saporin "specifically depletes host HSCs". As CD117 (c-kit) is also expressed on other stem and progenitor cells, the authors should consider rewording this. Additionally, this point should be explicitly mentioned and discussed, e.g. c-kit expression in germ stem cells and melanocytes. It is encouraging that the authors write "As CD117 is also expressed on a variety of other rare cell types throughout the body, full toxicity assessment was performed after CD117-ADC treatment." However, this reviewer feels that some more information on this topic should be given, e.g. in the discussion.*

Response: Based upon our experimental data showing depletion of HSC but preservation of other CD117 expressing progenitors, we feel we appropriately state that CD117-Saporin "specifically depletes host HSCs." However, to explain this better and address the point raised by the reviewer, we added the additional highlighted sentences to page 11 of the Discussion: "Interestingly, although CD117 is expressed on other rare cells throughout the body, such as germ cells and melanocytes, no gross toxicity was observed in these tissues after this CD117-ADC administration. Moreover, post this treatment many CD117 expressing progenitors were also surprisingly spared in the bone marrow despite robust HSC depletion."

Major point (#2): *In the view of this reviewer the comments by reviewers 1 and 3 have been adequately addressed.*

Response: We appreciate the reviewer's assessment of our revisions and similarly agree we have addressed the prior concerns wherever reasonably possible. These changes were noted in the prior point-by-point responses provided in our Nature Medicine re-submission, and we can re-provide them again if they would be helpful to further illustrate how we have responded to each of the reviewers' comments and concerns.

REVIEWER #1:

Remarks to the Author: *The authors have not performed a single experiment to answer my comments. The problem of small numbers of mice per group was not addressed. The hepatotoxicity was termed "no ideal" without any attempt to address this concern in more detail. Important controls are missing as also stated by the authors: "This was not tested as unfortunately we could not find an animal facility for which we could perform irradiation in virally infected animals." In my view this manuscript does not meet the quality criteria for Nature Communications.*

Response:

We respectfully disagree with this reviewer and would argue that we have addressed the reviewer's prior concerns by appropriately adjusting the text, displaying the data in clearer ways, and have included additional experimental results as requested. Each of the reviewer's past concerns was explained in the prior point-by-point responses that were previously provided to Nature Medicine, although we are not certain this was re-provided to the reviewer in this Nature Communication submission (we could re-send this if helpful). Included in this was the explanation that in total we have performed these types of transplants in repeated experiments with > 30 animals and are confident of our findings. Each experiment was only conducted with 4-5 mice/group due to the high costs of the reagents which prohibited larger cohorts, however all experiments were performed at least in duplicate. Additionally, due to the very large differences observed between experimental groups, we did not feel it was critical to increase the cohort sizes. We adjusted how we display this data by including data points of individual animals to better highlight the consistency of these results and have also now included statistical significance for each experiment to further illustrate these clear differences (in many cases $p < 0.0001$). Finally, we generated and included additional data to address the reviewers concerns and specifically added Supplemental Figure 4, which includes repeat experiments and additional controls. Additionally, we also conducted additional candida challenge experiments as requested by this reviewer to specifically quantify the effect on neutrophils and fungal load which were included in Figure 5h.

In regards, to the mild transient hepatotoxicity observed with the proof-of-concept CD117-ADC used in these studies, we explained and commented on this in detail in the discussion per the review's request specifically stating "As saporin toxin is known to have some hepatotoxicity which we similarly observed in our studies, alternative drug-conjugates may be preferred and utilized in similar fashions to further

improve the safety profile of such a conditioning approach. However, saporin may also be a viable ADC given it has been previously effectively used in clinical trials and in this type of conditioning setting would only need to be used as a one-time administration. Additionally, although some transient hepatotoxicity was seen with this CD117-ADC at the maximum 1.5mg/kg dose this is still likely a much-improved toxicity profile over current standard-of-care chemotherapy and/or irradiation conditioning.” Furthermore, we included additional studies showing efficacy of the CD117-ADC at reduced doses in Supplemental Figure 2g which should further have decreased hepatotoxicity, which enables an additional opportunity for non-toxic clinical translation of this agent. Given the known hepatotoxicity of saporin, we believe additional experiments investigating hepatotoxicity in this system are outside the scope of this manuscript and do not significantly add to the value of this body of work, especially as the CD117-ADCs utilized in these studies are proof-of-concept agents and may not be the ultimate clinical therapeutics used in patients. Based upon the findings in this manuscript, we are now developing further optimized CD117-antibody-drug-conjugates which we believe will have even more improved efficacy with minimal toxicity. We believe careful exploration of the toxicity of these optimized clinical-grade agents in mice and non-human primates is of much higher value and these experiments are currently underway: Pearse et al, ASH 2018. <https://ash.confex.com/ash/2018/webprogram/Paper114881.html>

Furthermore, we believe all appropriate controls were included in the manuscript and specifically we also included naked CD117 mAb in Figures 1c and Figure 1d at the reviewer’s request. We believe additional experiments would take considerable time given the long-nature of transplantation experiments, and we believe this important work should be shared expeditiously especially given its potential for clinical transplantation and its use as a tool to better study hematopoiesis. We believe this manuscript is of high-impact to both the scientific and clinical community and already have dozens of investigators that have contacted us to obtain and utilize these agents. Thus, we would appreciate this manuscript’s publication as is without further experimentation apart from that which we already included.

REVIEWER #2:

General comments: *This manuscript by Czechowicz et al. describes the HSC ablation using CD117-antibody-drug-conjugates enabling effective HSC transplantation with preservation of immunity. This is an interesting paper with a clear translational perspective, providing new options in conditioning during HSC transplantation.*

Response:

We appreciate the reviewer’s assessment of our work and the noted translatability of our approach. Given our findings, equivalent clinical agents are already in advanced clinical development (Hartigan, et al ASH 2017 and ASBMT 2018 Abstracts), and we are hopeful these will be quite meaningful therapeutics for patients and expand/enable improved hematopoietic cell transplantation. We believe such agents could be used broadly in treatment of non-malignant and malignant diseases, with recent evidence showing pre-clinical efficacy in both settings. Example studies:

Utility in Fanconi Anemia: Srikanthan et al ASH 2018:

<https://ash.confex.com/ash/2018/webprogram/Paper116851.html>

Utility in Acute Myeloid Leukemia: Proctor, et al ASH 2018:

<https://ash.confex.com/ash/2018/webprogram/Paper112726.html>

Major point (#1): *The authors state that CD117-Saporin "specifically depletes host HSCs". As CD117 (c-kit) is also expressed on other stem and progenitor cells, the authors should consider rewording this. Additionally, this point should be explicitly mentioned and discussed, e.g. c-kit expression in germ stem cells and melanocytes. It is encouraging that the authors write "As CD117 is also expressed on a variety of other rare cell types throughout the body, full toxicity assessment was performed after CD117-ADC treatment." However, this reviewer feels that some more information on this topic should be given, e.g. in the discussion.*

Response:

Based upon our experimental data showing depletion of HSC but preservation of other CD117 expressing progenitors (Figure 1 and Supplemental Figure 1), we feel we appropriately state that CD117-Saporin "specifically depletes host HSCs." However, to explain this better and address the point raised by the reviewer, as recommended we added the additional highlighted sentences to page 11 of the Discussion: "Interestingly, although CD117 is expressed on other rare cells throughout the body, such as germ cells and melanocytes, no toxicity was documented in these tissues after this CD117-ADC administration unlike after alternative conditioning treatments. Moreover, post this treatment many CD117 expressing hematopoietic progenitors were also surprisingly spared in the bone marrow despite robust HSC depletion, further implicating that not all CD117 expressing cells are killed by this CD117-ADC treatment which likely further contributes to its improved toxicity profile."

Additionally, we modified the toxicity data in the Results section to further point out the decreased toxicity of our CD117-ADC treatment compared to alternative methods and specifically in the revised manuscript now state on Page 8: "Even at the highest 1.5mg/kg CD117-ADC dose, this treatment was well-tolerated and non-myeloablative; moreover, treatment permitted long-term survival (>6 months) without requiring hematopoietic cell transplantation or transfusions unlike alternative treatments such as irradiation conditioning or ACK2 with CD47 blockade which has been reported to cause significant anemia. As CD117 is also expressed on a variety of other rare cell types throughout the body such as melanocytes and germ cells, toxicity assessment including full necropsy was performed after CD117-ADC treatment. Unlike post alternative conditioning strategies such as TBI or ACK2, recipients appeared healthy with full coat color and remained fertile with vigor (Fig S5a). At the highest 1.5 mg/kg optimal treatment dose, CD117-ADC conditioning did result in mild liver toxicity with transiently elevated transaminases and rare apoptotic hepatocytes (Fig S5b), and at increased doses animal mortality was intermittently observed. However, transaminitis resolved without intervention and no other toxicity was documented at this optimal dosing. Upon review by an experienced pathologist, all other organs examined had normal histology post treatment (Fig S5c)." We believe this additional information and the comparison to prior methods addresses the reviewer's request for comment and highlights the unique strengths of this CD117-ADC approach, and specifically the lack of graying of fur post CD117-ADC suggests melanocyte toxicity is likely lower compared to competitive antagonistic anti-CD117 antibody approaches (Fig S5a).

Notably to further explore the effects of the CD117-ADC in more detail on additional CD117+ cells, we did attempt staining of various mouse tissues post anti-mouse CD117-ADC treatment. Unfortunately, while we observed many preserved CD117+ cells on histology of various tissues, despite significant troubleshooting there was also troubling background staining of the anti-mouse CD117 antibodies used for immunohistochemistry and thus we do not feel confident in including these results in this manuscript. However, by flow cytometry we do show that in the bone marrow many CD117+ cells remain post CD117-ADC treatment (Supplemental Figure 1) despite near complete phenotypic and functional HSC depletion (Figure 1) and we predict similar findings in other tissues given the lack of

observed gross toxicity or histological toxicity even at long-term time points post CD117-ADC treatment (up to 24 weeks) (Supplemental Figure 5). However, despite this lack of observed toxicity, further detailed assessment will be critical prior to the initiation of clinical trials with similar clinical-grade anti-human CD117-ADCs. Such work is already underway with optimized clinical-grade anti-human CD117-ADCs, which thus far shows similar efficacy and minimal adverse side effects in non-human primate (NHP) models. We believe further experiments in this model will be more meaningful to confirm targeting CD117 with ADCs does not induce significant toxicities:

Pearse et al, ASH 2018. <https://ash.confex.com/ash/2018/webprogram/Paper114881.html>

We believe this is a much better model to explore and understand the potential toxicity of CD117-ADCs, and thus we feel that further toxicity work with the proof-of-concept 2B8-Saporin CD117-ADC in mice is of low additional value.

Major point (#2): *In the view of this reviewer the comments by reviewers 1 and 3 have been adequately addressed.*

Response:

We appreciate the reviewer's assessment of our revisions and similarly agree we have addressed the prior concerns wherever reasonably possible. These changes were noted in the prior point-by-point responses provided in our Nature Medicine re-submission, and we can re-provide them again if they would be helpful to further illustrate how we have responded to each of the reviewers' comments and concerns.